# Effects of β-Glucan Supplementation on LPS-Induced Endotoxemia in Horses

**DOI:** 10.3390/ani14030474

**Published:** 2024-01-31

**Authors:** Milena Domingues Lacerenza, Júlia de Assis Arantes, Gustavo Morandini Reginato, Danielle Passarelli, Júlio César de Carvalho Balieiro, Andressa Rodrigues Amaral, Thiago Henrique Annibale Vendramini, Marcio Antonio Brunetto, Renata Gebara Sampaio Dória

**Affiliations:** 1Department of Veterinary Medicine, Faculty of Animal Science and Food Engineering, University of Sao Paulo (USP), Pirassununga 13635-900, Brazil; milena.lacerenza@usp.br (M.D.L.); julia.arantes@usp.br (J.d.A.A.); gmorandinivet@gmail.com (G.M.R.); passarelli.dani@usp.br (D.P.); 2Pet Nutrology Research Center, Nutrition and Production Department, School of Veterinary Medicine and Animal Science, University of Sao Paulo (USP), Pirassununga 13635-900, Brazil; balieiro@usp.br (J.C.d.C.B.); andressa.rodrigues.amaral@usp.br (A.R.A.); thiago_vendramini@hotmail.com (T.H.A.V.); mabrunetto@usp.br (M.A.B.)

**Keywords:** cytokines, horses, immunomodulation, lipopolysaccharides, yeasts

## Abstract

**Simple Summary:**

Endotoxemia in horses is a serious condition caused by diseases that result in systemic inflammation, such as colic, pleuropneumonia, metritis, etc., and can lead to death. Finding a treatment or prevention for this condition is of great importance for the health of horses. The β-glucans present in yeast, especially *Saccharomyces cerevisia*, are supplemented with nutritional properties that induce the modulation of the immune system. This study aims to evaluate whether supplementing horses’ diets with β-glucans is capable of modulating horses’ immune response to the inflammatory stimulus caused by the intravenous injection of endotoxins (*E. coli* lipopolysaccharide; 0.1 µg/kg/body weight). It was found that there is a positive interference with this supplementation, with evidence of the modulation of the immune system, which encourages its use as a dietary supplement in order to assist the immune response of horses in cases of endotoxemia.

**Abstract:**

β-glucan is part of the cell wall of fungi and yeasts and has been known for decades to have immunomodulating effects on boosting immunity against various infections as a pathogen-associated molecular pattern that is able to modify biological responses. β-glucan has been used in rat models and in vitro studies involving sepsis and SIRS with good results, but this supplement has not been evaluated in the treatment of endotoxemia in horses. This study aims to evaluate the effects of preventive supplementation with β-glucan in horses submitted to endotoxemia by means of inflammatory response modulation. Eight healthy horses, both male and female, aged 18 ± 3 months, weighing 300 ± 100 kg of mixed breed, were randomly assigned to two groups of four animals, both of which were subjected to the induction of endotoxemia via the intravenous administration of *E. coli* lipopolysaccharides (0.1 µg/kg). For 30 days before the induction of endotoxemia, horses in the β-glucan group (GB) received 10 mg/kg/day of β-glucan orally, and horses in the control group (GC) received 10 mg/kg/day of 0.9% sodium chloride orally. The horses were submitted to physical exams, including a hematological, serum biochemistry, and peritoneal fluid evaluation, and the serum quantification of cytokines TNF-α, IL-6, IL-8, and IL-10. For statistical analysis, the normality of residues and homogeneity of variances were verified; then, the variables were analyzed as repeated measures over time, checking the effect of treatment, time, and the interaction between time and treatment. Finally, the averages were compared using Tukey’s test at a significance level of 5%. Horses from both experimental groups presented clinical signs and hematological changes in endotoxemia, including an increase in heart rate and body temperature, neutrophilic leukopenia, an increase in serum bilirubin, glucose, lactate, and an increase in TNF-α, IL-6, and IL-10. Hepatic and renal function were not compromised by β-glucan supplementation. GB presented higher mean values of the serum total protein, globulins, and IL-8 compared to that observed in GC. In the peritoneal fluid, horses from GB presented a lower mean concentration of neutrophils and a higher mean concentration of macrophages compared to the GC. It was concluded that preventive supplementation of β-glucan for thirty days modulated the immune response, as evidenced by increasing serum total proteins, globulins, IL-8, and changes in the type of peritoneal inflammatory cells, without effectively attenuating clinical signs of endotoxemia in horses. Considering the safety of β-glucan in this study, the results suggest the potential clinical implication of β-glucan for prophylactic use in horse endotoxemia.

## 1. Introduction

The development of clinical endotoxemia in horses is frequent as a result of the high sensitivity of this species to the lipopolysaccharides (LPSs) of the external membrane of Gram-negative bacteria, which are released into the circulation during the septic shock process [1]. In response to systemic LPS, the organism releases pro-inflammatory substances, cytokines, into the circulatory system [2]. This condition is present in most cases of the equine acute abdomen as a consequence of the systemic inflammatory response syndrome, resulting in complications like jugular thrombophlebitis, laminitis, and adynamic ileum, among others, decreasing the short and long-term survival rates of horses with this disease [3,4,5].

The search for greater success rates in the treatment of animals affected by equine endotoxemia, associated with the possibility of using by-products from the alcohol industry, has stimulated studies on β-glucans, which have therapeutic properties, important nutritional potential and have been intensively researched and incorporated into human and animal nutrition (pigs, dogs, cats, rats and fish) with interesting responses: modified immune responses, a reduction in inflammation responses, and altered glucose and lipid metabolism [6,7]. The current market asks for alternative and natural products that promote new studies that use non-drug treatment options.

β-1→3-D-glucans occur as a major component of microbial cell walls or can be secreted by both non-pathogenic and pathogenic fungi such as *Saccharomyces cerevisiae* [8]. These β-1→3-D-linked glucose polymers are characterized as fungal pathogen-associated molecular patterns (PAMPs) and have the ability to modify biological responses [9].

Currently, β-glucan from yeast has been used extensively as a protective substance against infections due to their ability to stimulate the immune system, with potent effects on innate and adaptive immunity [10,11]. These polymers activate the immune response via the complement system, either directly or with the aid of antibodies, and produce chemotactic factors, which induce the migration of leukocytes to the infection site, thus presenting immunomodulatory activity. It has also been suggested that β-glucan can directly activate leukocytes, stimulating their phagocytic, cytotoxic functions and antimicrobial activity, in addition to stimulating the production of pro-inflammatory mediators [6]. The protective effect of β-glucan has been demonstrated in the in vitro and in vivo experimental studies involving Leishmania major and *L. donovani*, *Candida albicans*, *Toxoplasma gondii*, *Streptococcus suis*, *Plasmodium berghei*, *Staphylococcus aureus*, *Escherichia coli*, *Mesocestoides corti*, *Trypanosoma cruzi*, *Eimeria vermiformis* and *Bacillus anthracis* [12,13].

In addition, β-glucan seems to be able to modify the response to pro-inflammatory stimuli or even sepsis. In a murine polymicrobial sepsis model, β-glucan treatment resulted in decreased septic morbidity and mortality, demonstrating the protective role of β-glucan in certain pro-inflammatory conditions [14,15,16]. The mechanisms underlying these beneficial effects involve the fact that, in the presence of a pro-inflammatory response, β-glucan has been shown to strongly induce the interleukin-1 antagonist receptor (IL-1 RA), suggesting the immediate anti-inflammatory potential of β-glucan, signaled by switching a pro- to an anti-inflammatory IL-1RA-mediated reaction [17].

β-glucan has been used in rat models and in vitro studies of sepsis and SIRS with good results, but this supplement was not evaluated for the treatment of endotoxemia in horses that we could find in the literature consulted. Thus, it is expected that β-glucan from yeast, which has the modulating activity of systemic immunity, may be beneficial in the control of endotoxemia in horses, modulating the systemic inflammatory response. Therefore, the objectives of this study were to evaluate whether the preventive use of β-glucan orally influences physical parameters, hematological, serum biochemical, and peritoneal fluid values, and the serum expression of inflammatory cytokines in horses with induced endotoxemia.

## 2. Materials and Methods

The research project was approved by the Animal Experimentation Ethics Committee of FZEA/USP (protocol nº 3938210218).

Eight healthy, mixed-bred horses, both male (*n* = 4) and female (*n* = 4), aged 18 ± 3 months and weighing 300 kg ± 100 kg, were used in this study. The sample size was a convenience sample based on some previous studies. The horses had no recent history of gastrointestinal illness and were considered clinically healthy based on a physical examination and routine hematological and biochemical blood profiles. Horses were housed indoors throughout the experimental period and fed a maintenance diet consisting of Bermuda grass (*Cynodon dactylon*) hay (2% of body weight) and commercial horse feed (1% of body weight). Water and trace-mineral salt were provided ad libitum. The horses were not exercised during the study period but had free access to a paddock once a day for thirty minutes. Thirty days before the induction of endotoxemia, two female and two male horses received 10 mg/kg of β-glucan orally daily (group β-glucan; GB), and two females and two male horses received 10 mg/kg of 0.9% sodium chloride, orally daily (control group; CG), which was selected as the placebo. β-glucan and sodium chloride were weighed daily and administered manually with a syringe directly into the mouth of the animals, confirming the consumption and the amount administered. Purified β-1,3/1,6-glucan extracted from the cell wall of the brewer yeast *Saccharomyces cerevisiae* was used in this study. A biotechnological process of enzymatic isolation was used to remove undesired cell layers (i.e., chitin, mannanoligosaccharides, and a low portion of mannoproteins) and to obtain highly purified and bioactive β-1,3/1,6-glucans.

In all horses, sublethal endotoxemia was induced via the intravenous (IV) administration of 0.1 µg/kg of *E. coli* 055: B53 endotoxin, administered in 250 mL of 0.9% sodium chloride solution for 15 min. The moments of evaluation of the animals regarding physical examination, blood sampling, and the collection of peritoneal fluid occurred immediately before induction of endotoxemia (T0), immediately (T0+), and 15 (T15′), 30 (T30′), 60 (T60′), 90 (T90′), 120 (T2h), 180 (T3h), 240 (T4h), 360 (T6h), 480 (T8h), 600 (T10h) and 720 (T12h) minutes after the induction of endotoxemia.

The horses were evaluated for heart rate (HR; beats per minute; bpm), respiratory rate (RR; respiratory movements per minute), and rectal temperature measurement (°C). Blood was evaluated for packed cell volume (PCV), red blood cell counts (RBCs), total and differential white blood cell counts (WBCs and WBCs differential), and platelet counts. Biochemical analyses measured the hepatic function (AST—aspartate aminotransferase; GGT—gamma-glutamyl transferase; total proteins; albumin; globulins; fibrinogen; DB—direct; IB—indirect; and TB—total bilirubin), renal function (urea and creatinine), serum glucose and lactate. Peritoneal fluid was monitored by means of a visual examination (appearance and color), density, coagulation, occult blood, and pH; cytological analysis was conducted by means of the red blood cell count, total nucleated cell count, and leukocyte differential count; and biochemical parameters were measured including the total protein, glucose, lactate. and fibrinogen. All samples (blood and peritoneal fluid) were analyzed using the same automated analyzer (Mindray BC-2800 Vet^®^, Mindray Medical International Limited, Shenzhen, China). Blood and peritoneal fluid lactate and glucose concentrations were determined immediately after collection using Accutrend Plus (Roche Diagnostics, São Paulo, Brazil) and OptiumXido (Abbott Diabetes Care, São Paulo, Brazil), respectively. A rapid heat precipitation micro method was used for the estimation of plasma or peritoneal fluid fibrinogen. Data were manually recorded at each measurement time point.

Immediately after collection, 500 µL aliquots of blood serum were stored in cryogenic tubes, frozen at −10 °C and, subsequently, in a freezer at −80 °C, for the measurement of TNF-α, IL-6, IL-8, and IL-10 using ELISA enzyme tests with commercial kits (eBIOSCIENCE^®^, San Diego, CA, USA), according to the manufacturer’s instructions (see Appendix A). Two replicates of each sample and standards per ELISA assay were used.

A general mixed linear model with the fixed effects of treatment (control and β-glucan), time (T0, T0+, T15′, T30′, T60′, T90′, T2h, T3h, T4h, T6h, T8h, T10h, and T12h), the interaction treatment × time, and random effects of the animal and residue was used for the statistical analysis of the results. In these analyses, the structure of repeated time measurements in the same experimental units was assumed. The covariance structures between repeated measurements were assessed using the Akaike Information Criterion (AIC) [18]. The assumptions of the analysis of variance models were simultaneously tested via studentized conditional residual analyses. For time comparisons within each treatment, Tukey’s test was used as a method for the mean comparison at a significance level of 5%.

## 3. Results

Preventive β-glucan supplementation did not influence the clinical signs of horse endotoxemia. There was an observed increase in heart rate and an increase in rectal temperature in both experimental groups over the 12 h evaluated. The mean HR and body temperature increased from T0+ (43 ± 4 bpm and 37.3 ± 0.8 °C), reaching a HR peak of 57 ± 9 bpm in T2h and a body temperature peak of 39.1 ± 0.7 °C in T4h, which was maintained above T0 (42 ± 4 bpm and 36.9 ± 0.8 °C) until T12h (54 ± 8 bpm and 38.3 ± 0.5 °C).

There was an observed decrease in WBC counts during the evaluative period, represented by neutrophilic leukopenia, in both experimental groups, with no differences between groups. The decrease in WBC counts started from T0+ (6.7 ± 1.96 × 10^3^ leukocytes/µL), with a peak at T90′ (2.58 ± 0.74 × 10^3^ leukocytes/µL). Leukopenia is maintained for T4h, with an increase in mean WBC counts starting at T6h (5.49 ± 2.42 × 10^3^ leukocytes/µL), although this was at concentrations even lower than T0 (7.81 ± 1.69 × 10^3^ leukocytes/µL). At T8h (7.33 ± 2.53 × 10^3^ leukocytes/µL), the values returned to what was considered physiological for the species, whereas at T10h (8.40 ± 2.50 × 10^3^ leukocytes/µL) and T12h (9.05 ± 2.43 × 10^3^ leukocytes/µL) there was an increase in the average concentration of WBC, in relation to T0, although values were within the physiological range for the species (5.2 to 13.9 × 10^3^ leukocytes/µL) [19].

An increase in total and indirect bilirubin and glucose was observed; however, the values remained within the considered physiological range for the species (TB: 0.5 to 2.1 mg/dL; IB: 0.2 to 2 mg/dL; Glucose: 75 to 115 mg/dL) [19], and there were no differences in these parameters between the experimental groups. The increase in bilirubin started from T0+ (TB: 1.02 ± 0.47 mg/dL; IB: 0.7 ± 0.53 mg/dL), with a peak at T8h (TB: 1.8 ± 0.3 mg/dL; IB: 1.58 ± 0.32 mg/dL), remaining above T0 (TB: 0.8 ± 0.49 mg/dL; IB: 0.47 ± 0.49 mg/dL) until T12h (TB: 1.40 ± 0.66 mg/dL; IB: 1.11 ± 0.67 mg/dL) and glucose, respectively, with elevation (T0: 101.46 ± 7.99 mg/dL) starting at T60′ (108.9 ± 22, 44 mg/dL), with a peak at T12h (117.75 ± 9.63 mg/dL). An increase in serum lactate was observed, starting at T15′ (1.43 ± 0.45 mmol/L), remaining above T0 (1.04 ± 0.29 mmol/L) until T12h (1.69 ± 0.37 mmol/L), and reaching values above the physiological for this species (serum lactate: 1.11 to 1.78 mmol/L) [19] at T8h (2 ± 0.49 mmol/L) and T10h (1.88 ± 0.31 mmol/L).

Considering the differences between groups, GB presented higher averages of serum AST, total proteins, globulins and lactate and lower averages of GGT compared to GC (Table 1), but all values remained within what is considered physiological for the species, except the globulins that presented average values above the reference values in GB.

In the peritoneal fluid, an increase in lactate was demonstrated over time, with a peak at T8h (1.67 ± 0.41 mmol/L), remaining above T0 (0.65 ± 0.25 mmol/L) and presenting values above those considered physiological for the species (0.39 to 1.19 mmol/L) [18], from T60′ (1.23 ± 0.36 mmol/L) to T12h (1.23 ± 0.51 mmol/L). Comparing between groups, GB presented a higher average of peritoneal lactate compared to GC (Table 2), with the ratio of peritoneal fluid with plasma lactate reaching 0.75 in GC and 0.85 in GB. No changes regarding total nucleated cell counts were observed in the peritoneal fluid; however, there was a difference between groups in the percentage of the type of inflammatory cells. GB presented with a lower percentage of segmented neutrophils and a higher average of macrophages in peritoneal fluid than the GC (Table 2).

Regarding inflammatory cytokines, there were changes in the mean concentrations of TNF-α (CV = 0% and R^2^ = 1), IL-6 (CV = 1.24% and R^2^ = 1), and IL-10 (CV = 1.91% and R^2^ = 1) over time, from T0 + to T12h in both experimental groups. Comparing between groups, GB presented higher means of IL-8 (CV = 0.25% and R^2^ = 1) than GC (Table 3). For IL-6, there were two peaks in the mean concentration at T30 (194.44 ± 452.48 pg/mL) and T3h (244.35 ± 165.20 pg/mL), followed by a reduction in mean serum concentrations, although there were values still above T0 (22.23 ± 33.58 pg/mL), even at T12h (47.90 ± 56.17 pg/mL) (Figure 1A). With regard to IL-8, a peak was found at T30′ (664.83 ± 1012.33 pg/mL), with a subsequent reduction in concentrations over time, from T60′ (174.24 ± 110.91 pg/mL) to T12h (154.69 ± 119.13 pg/mL), with mean values below T0 (212.35 ± 177.23 pg/mL) (Figure 1B). Regarding IL-10, there was an increase over time, starting at T15′ (193.55 ± 182.08 pg/mL), with a peak at T4h (896.5 ± 317.7 pg/mL), which remained at a plateau of up to T6h (893.63 ± 306.47 pg/mL), with a subsequent reduction until T12h (270.43 ± 119.37 pg/mL), although there were values above T0 (139.48 ± 139.79 pg/mL) (Figure 1C). TNF-α showed an increase in serum concentration averages over time, with a peak at T90′ (2785.13 ± 1495.63 pg/mL) and a subsequent continuous reduction over time, although, from T6h (32.43 ± 24.14 pg/mL) to T12h (13.77 ± 23.83 pg/mL), the values were still above T0 (6.54 ± 9.22 pg/mL) (Figure 1D, Appendix A).

No changes were demonstrated over time and between groups for the other parameters evaluated.

## 4. Discussion

This study is a pioneer in the evaluation of the preventive inclusion of β-glucan in the diet of horses, aiming at an immunomodulatory effect. Based on the findings of this present study, oral supplementation, with 10 mg/kg/day of β-glucan, appears safe and presents results that are consistent with immunological modulation in horses with induced endotoxemia without effectively interfering in the clinical signs and hematological changes promoted by induced endotoxemia in horses.

The evaluation of horses over time showed a picture of systemic endotoxemia, illustrated by an increase in heart rate and in body temperature and neutrophilic leukopenia for 4 h, followed by a compensatory increase in WBC (neutrophils) after 10 h of LPS administration [1,21,22,23]. This leukopenia usually occurs due to the sequestration of neutrophils in the microcirculation in a transitory fashion once LPS-induced endotoxemia, in a sub-lethal dose, promotes transient endotoxemic clinical signs and hematological changes. The higher concentrations of serum and peritoneal lactate, after LPS administration, in the group supplemented with β-glucan compared to the control group, reaching mean peritoneal values slightly above the physiological for this species, is consistent with the intestinal hypoperfusion and abnormal microvascular control of oxygenation induced via endotoxemia [24,25]. However, the ratio of peritoneal fluid/plasma lactate was lower than one in both groups, which is considered clinically normal for horses [19]. It is hypothesized that preventive supplementation with β-glucan did not interfere with or promote any benefit in vascular perfusion and oxygen supply to the tissues during endotoxemia in horses [26]. The difference observed between the groups probably reflects the individual response of the horses to the intravenous sublethal LPS challenge.

Preventive supplementation with β-glucan resulted in a higher average concentration of total serum proteins and globulins compared to the control group. These findings suggest a better immune response of β-glucan supplemented horses since globulins, especially immunoglobulins, are directly involved with antigen binding and the biological immune response following antigen binding [27].

β-glucan supplementation promoted an alteration in the cell type present in peritoneal fluid, presenting with a lower concentration of neutrophils and higher concentration of macrophages than the group control, without changing the total nucleated cell count. It is adequately documented that endotoxemia induced by the intravenous administration of LPS does not change the total number of white blood cells in peritoneal fluid [28]. Although peritoneal changes in response to chemotactic inflammatory mediators, such as the ones induced by intestinal ischemia and/or peritonitis, include an increase in the number of neutrophils that enter the peritoneal cavity [29]. The differences in peritoneal cellularity observed between groups in this study suggest that preventive supplementation with β-glucan may modulate the biological response, triggering a series of events in the immune response in horses and resulting in the alteration of inflammatory cell types in peritoneal fluid during induced endotoxemia.

Several inflammatory mediators, derived from the host cells, are responsible for most manifestations of endotoxemia, causing an increase in the production of endogenous cytokines, such as TNF-α and IL-6, IL-8, and IL-10, which play a critical role in inflammatory responses [30,31]. The serum concentration of TNF-α rises 15 to 30 min after the induction of endotoxemia, reaching a maximum peak of around 60 to 90 min [1,30,31,32,33,34]. Pro-inflammatory cytokine IL-6, when activated via LPS, with peaks at 30 min and 3 h, promotes a series of genetic, biochemical, and clinical events. This cytokine acts on the release of secondary mediators, which are responsible for reactivating phagocytic cells and the inflammatory cascade [24,35]. Interleukin-10 (IL-10) is an important anti-inflammatory cytokine in the pathophysiology of sepsis, which seeks to counteract the actions of pro-inflammatory mediators by reducing the synthesis and release of these mediators while antagonizing their effects [36]. In this study, there was a clear increase, over time, in TNF-α, IL-6, and IL-10 in both experimental groups, which is characteristic and consistent with the correct induction of endotoxemia.

Interleukin 8 (IL-8) is a potent substance attractive to neutrophils. Its action is mediated by CXCR1 and CXCR2 receptors coupled to protein G (GPCRs), which have a high affinity for positive glutamine–leucine–arginine chemokines [37,38]. Endotoxins were able to increase the expression of IL-8 in neutrophils in vitro [36] and in the blood when injected systemically [39]. Considering that IL-8 is a key molecule for the migration and activation of neutrophils, the verification, in this study, that horses supplemented with β-glucan had higher mean concentrations of this cytokine in comparison to the control group demonstrates the possible immunomodulatory activity of the product since it may have stimulated the production of this pro-inflammatory mediator [6].

It is known that β-glucan has immunomodulating effects on boosting immunity against various microbial infections [12,40]. Moorlag et al. (2020) [41] showed that β-glucan induces protective and trained immunity in human monocytes and in mice infected with virulent Mycobacterium tuberculosis, and β-glucan-mediated protection is dependent on the interleukin-1 (IL-1) signaling pathway. β-glucan promotes epigenetic and functional changes in innate defense, resulting in the genetic reprogramming of innate mononuclear immune cells. In cases of sepsis and SIRS, the goal of treatment is active protective mechanisms that modulate the pro-inflammatory cell response, which attempts to convert a pro-inflammatory stimulus into an anti-inflammatory response [42].

β-glucan has been used in rat models and in vitro studies involving sepsis and SIRS, with good results once they convert the pro-inflammatory response to an anti-inflammatory response via the IL-1 antagonist receptor (IL-1 RA). LUHM et al. (2006) [17] provide evidence that downstream of the recognition and signaling pro-inflammatory transcription factor-binding and cytokine expression, human leukocytes are switched to an anti-inflammatory phenotype by β-glucan and become synergic via the up-regulation of IL-1 RA. Also, it demonstrated a positive correlation between the elevation of IL-8 with the elevation of IL-1 RA. Flow cytometric experiments confirm that monocytes and neutrophils are able to produce IL-8, such as IL-1 RA, in response to β-glucan. Altogether, the induction of the neutrophil-attracting IL-8 and the anti-inflammatory IL-1RA via fungal carbohydrates (β-glucan) may well fit a benign pathogen-associated molecular pathogen (PAMP) response, mounting defensive mechanisms against a possible microbial attack [17]. Therefore, it is hypothesized that the increase in IL-8 observed in this study may represent the immunomodulatory effect of the inclusion of β-glucan in the diet of horses. Further studies should be conducted to determine the mechanisms of action and establish the ideal dosage of supplementation for equine species to achieve the immunomodulation of inflammatory response in equine endotoxemia.

The evaluation of enzymes that indicate hepatic and renal injury demonstrated that oral supplementation with 10 mg/kg of β-glucan for 30 days can be considered safe and does not present systemic adverse effects once the mean values were confirmed to remain within the considered physiological for this species [43,44].

The number of horses (sample size) used in this study was based on some previous studies by our research group aimed at evaluating the influence of β-glucan supplementation in other animal species. These studies showed significant effects [45,46].

Some limitations should be noted in the present study. First, the small number of healthy horses used makes it difficult to consider the results and support the use of β-glucan in clinical cases. Second, a dose–response study was not performed prior to this work. Third, IL-1β or IL-1RA were not measured in our study due to financial limitations. And fourth, the LPS model of endotoxemia induction in horses (transitory endotoxemia) probably differs from a clinical case of endotoxemia, sepsis, or SIRS in sick horses once it is known that different immune responses can be attributed to delicately balanced differences in signaling pathways between stimuli, such as LPS or another bacterial antigen [17].

Therefore, as the dose of the supplement used in this study was adapted for horses and since there are no studies previously performed on this species, it is assumed that further research should be conducted in order to assess the dose-response effects of this supplement, seeking to highlight its immunomodulatory activity. Further studies are needed to recommend the use of β-glucans in equine nutrition. This study is pioneering and brings important evaluations to light. However, in terms of safety, other body systems may also be affected by β-glucans, and so far, there is little information on the effects of its long-term use or use in other conditions (e.g., in sick animals), such as prophylactic use in equine endotoxemia.

## 5. Conclusions

The preventive oral supplementation of β-glucan at a dose of 10 mg/kg/day for thirty days modulates the immune response, as evidenced by an increase in serum total proteins, globulins, IL-8, and changes in the type of peritoneal inflammatory cells, without effectively attenuating the clinical signs of endotoxemia in horses.

## Figures and Tables

**Figure 1 animals-14-00474-f001:**
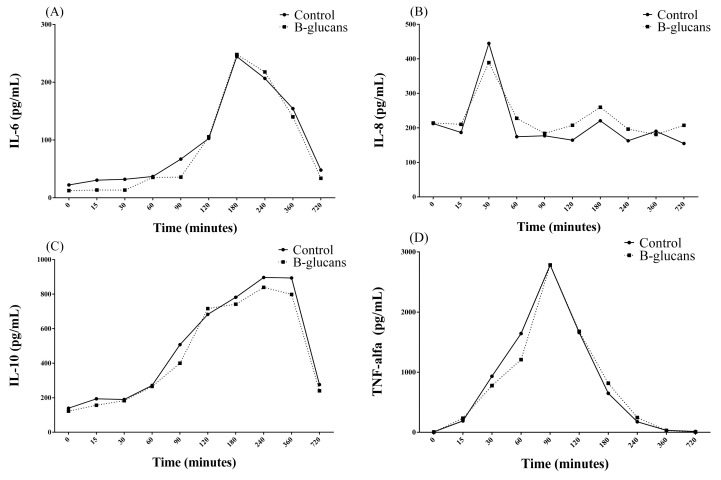
Changes in the mean concentrations of IL-6 (**A**), IL-8 (**B**), IL-10 (**C**) and TNF-α (**D**) over time in the control group and β-glucan group.

**Table 1 animals-14-00474-t001:** Serum biochemistry values as a function of time and experimental treatments in the control group and β-glucan group.

Variable	Group		SEM ^4^	*p*		
	Control	β-Glucans		Treatment	Time	Treatment × Time
AST ^1^ (U/L)	287.56	315.60	14.118	0.022	1.000	1.000
GGT ^2^ (U/L)	12.90	9.27	28.445	0.001	0.993	0.994
TP ^3^ (g/dL)	6.52	6.77	5.765	0.021	0.971	0.992
Globulins (g/dL)	3.72	4.01	9.937	0.010	0.991	0.988
Lactate (mmol/L)	1.40	1.64	30.451	0.026	0.001	0.953

^1^ AST: aspartate transaminase; ^2^ GGT: gamma-glutamyl aminotransferase; ^3^ TP: total proteins; ^4^ SEM, standard error of the mean. *p* ≤ 0.05. Reference values: AST: 226–366 U/L; GGT: 9.0–25.0 U/L; total proteins: 5.5–7.3 g/dL; globulins: 2.1–3.8 g/dL; lactate: 1.11–1.78 mmol/L [19,20].

**Table 2 animals-14-00474-t002:** Evaluation of peritoneal fluid as a function of time and experimental treatments in the control group and β-glucan group.

Variable	Group		SEM ^1^	*p*		
	Control	β-Glucans		Treatment	Time	Treatment × Time
Lactate (mmol/L)	1.05	1.40	0.413	0.003	<0.001	0.997
Neutrophils (%)	65.87	55.85	31.130	0.020	<0.001	0.852
Macrophages (%)	25.33	34.60	57.400	0.020	0.002	0.673

^1^ SEM, standard error of mean. Reference values: Lactate: 0.39–1.19 mmol/L; neutrophils: 36–78%; macrophages: 3–50% [19,20]. *p* ≤ 0.05.

**Table 3 animals-14-00474-t003:** Serum cytokine values as a function of time and experimental treatments in the control group and β-glucan group.

Variable	Group		SEM ^5^	*p*		
	Control	β-Glucans		Treatment	Time	Treatment × Time
IL-6 ^1^ (pg/mL)	99.06	122.39	156.422	0.549	0.05	0.499
IL-8 ^2^ (pg/mL)	156.24	305.20	141.847	0.05	0.078	0.091
IL-10 ^3^ (pg/mL)	518.81	526.88	82.869	0.923	<0.001	0.458
TNF-alfa ^4^ (pg/mL)	839.45	778.39	154.134	0.787	<0.001	0.989

^1^ IL-6, interleukin-6; ^2^ IL-8, interleukin-8; ^3^ IL-10, interleukin-10; ^4^ TNF-alfa, tumor necrosis factor alpha; ^5^ SEM, standard error of mean. *p* ≤ 0.05.

## Data Availability

Data supporting the reported results can be found at http://dedalus.usp.br and https://repositorio.usp.br/index.php. Available online: https://uspdigital.usp.br/siicusp/siicPublicacao.jsp?codmnu=7210 accessed on 28 January 2024.

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
