# Peer review of "Effects of β-Glucan Supplementation on LPS-Induced Endotoxemia in Horses"

_animals, 2024, doi:10.3390/ani14030474_

Round 1

Reviewer 1 Report

Comments and Suggestions for Authors

Review of Lacerenza et al.

General comments: this is an interesting paper covering a novel topic. The authors aim to modulate endotoxic effects using Beta glucans in a simulated model. Overall the paper presents interesting data that are valuable additions to the literature. I think the paper could benefit from some English language editing, although the mistakes are fairly minor. Secondly, I think their statistics section needs to be lengthened in order to understand how they analyzed their data. Did they use repeated measures? What was the repeated term, which covariance matrix was used? I often find that when analyzing cytokines, that the data need to be transformed for normality or that time 0 covariates are needed. If the authors are not comfortable with this, I would suggest they consult a statistician that is fluent in repeated measures analyses. Third- the tables need to be modified, as there is no presentation of the time lsmeans. For all data in the tables, we need the error to be added (either SEM or Confidence Intervals). Presenting some as figures might be helpful as well. Fourth- I think the citations are in the correct format, please check the journal guidelines.

Specific comments:

Line 58: this sentence does not flow, please revise. Unsure on what the authors are meaning here.

Lines 63-68: this is very generic information. I think the reader needs a more detailed introduction to why the hypothesis was developed. Which species are included in “animal nutrition”.  “The current market appeal is for alternative..”

Line 71: remove comma

Line 93: “shown to strongly”

Line 97: authors state there is no research. Always like to justify with “that we could find”. Also there is a variety of related literature using beta glucans in horses. Plenty that cover immunological responses to other inflammatory mediators. So I would rephrase this to be less severe.

Line 143: There was an observed increase

Line 148: same as above

Line 157: Physiological range for the species

Line 160: same as above

Line 186: there was a difference between

Line 224-237: inadequate discussion of the lactate response. The discussion is superficial. Please do further reading to find a hypothesis about why lactate was higher in beta glucans. Do horses ferment them to lactate?

Line 237: hypothesized

Line 246: do not being a paragraph with “also”

Line 273-275: while tempting to interpret numerical differences, this should be deleted. Or do your stats with a statistician that knows how to reduce variation (covariates are helpful here). Reassess this after looking at stats. If not significant, then remove section.

Line 301- remove “of”

Comments on the Quality of English Language

Minor edits needed.

Reviewer 2 Report

Comments and Suggestions for Authors

Main comments:

This manuscript has some English grammar issues and should thus undergo careful language review. In addition, the conclusion should be adjusted as it is currently not supported by the results of this study. There are also multiple open questions concerning the methodology of this study and the results are not easy to follow. The latter could be resolved through the inclusion of figures illustrating the results. Please see my detailed comments and questions regarding this manuscript below.

Detailed comments:

Abstract:

-          The conclusion should be adjusted. I think that based on your findings there is currently no reason to suggest that this supplement should be used. That’s because it does not seem to positively alter clinical signs. Also, while you found changes in the immune responses, it is unclear whether these are beneficial or not.

Introduction:

-          Based on the putative immune-activating function of beta-glucans discussed in the introduction, it seems to me that beta-glucans could cause unnecessary inflammation and bystander tissue-damage, if fed preventatively (i.e., in the absence of a medical condition). Please consider commenting on this. In line 73 and till the end of the paragraph: Please consider adding information about the study design of these previous studies. Did they also administer a preventative dose for 30 days? Also, please consider inclusion of a statement about long-term side effects.

Methods:

-          How did you determine the health status of the horses?

-          Why did you use 8 horses? Did you perform a sample size calculation, or was this a convenience-sample? If the latter is true, please state that explicitly, and discuss the implications of this approach. 

-          Please provide more information on how the horses were fed, housed, and exercised during the study. Were these parameters standardized? Were horses kept on pasture? Did they receive any other supplements, therapies, or concentrated feeds?

-          Please explain why sodium chloride was chosen as a control. Is this a vehicle control? Also, where was the beta-glucan obtained and how was its purity ensured?

-          How was the beta-glucan exactly provided to the horses? Top-dressed over their grain/syringed with water? How did you ensure that all beta-glucan was received?

-          For all analyses performed (WBC etc.), please provide information about the machine(s) used.

-          For the ELISA assays:  Please provide information on intra- and inter-assay reliability/CVs, the assay sensitivity, the lower and upper detection limits and the curve fit R2.

Results:

-          I have a hard time following your results section because it is so data-dense. I recommend presenting all your results in figures and explaining only significant differences in the text.

-          Line 149: Is a “respectively” missing after glucose?

-          Line 150 (and in following lines): What does BT and BI stand for?

Discussion and conclusion:

-          Throughout your discussion and conclusion sections, please reconsider statements that suggest that this supplement should be recommended for use. Concerning safety, you studied hepatic and renal function only, but other body systems may be affected by beta-glucans as well, and you do not have any information regarding effects of longer-term use or use in “unhealthy horses” etc. (your sample size was very small). Thus, I am not sure that beta-glucans should be considered totally safe for use in horses at this point. In addition, since the clinical signs were not improved, I think that we can currently not believe that this is a useful supplement.

-          Paragraph starting in line 230: I wonder if supraphysiological antibody concentrations, which the beta-glycan appears to induce, may increase the risk of autoimmunity, cell bystander-damage, the development of inflamm-aging, and longer-term reduction in immunological competence. Maybe this supplement should not be fed over a longer period of time/preventatively. Do you have any insights in this regard that should be shared with the reader?

-          Throughout your discussion I feel like you summarize the literature a lot without direct connection to your findings. Maybe you could make the discussion a bit more concise?

-          Why did you not measure IL-1b or IL-1RA in your study? Especially IL-1RA appears to be very relevant to this work. R and D systems offer a good ELISA assay for equine IL-1RA.

-          Please explain in the methods what you considered a ‘trend’/ a “tendency” in terms of the p-values, and in the discussion, please do not over interpret non-significant results that appear to be ‘a trend’.

-          Why was a dose-response study not performed prior to this work?

Comments on the Quality of English Language

This manuscript has some English grammar issues and should thus undergo careful language review.

Round 2

Reviewer 1 Report

Comments and Suggestions for Authors

Authors have done a good job of including additional information and interpretation. 

Author Response

Dear Reviewer,

The authors wish to thank you for the attention given to our research. Thank you once again for considering our work for publishing.

Reviewer 2 Report

Comments and Suggestions for Authors

The authors addressed many of my concerns, and I think that the results section has been significantly improved. Unfortunately, there are still language issues in this manuscript, particularly in sections added during the last review process. I have provided examples of language issues below.

I appreciate that you answered my question about how you arrived at n= 8 horses for this study, but I think it would be best to add this explanation to the article as well. Similarly, I appreciate that you explain to me why sodium chloride was selected as the control/placebo, but I think it would be best if this information was also provided in the manuscript. Regarding your supplemental file with the ELISA data: unfortunately, this was not provided to the reviewers, but I understand that it contains information such as sensitivity and upper and lower detection limits. What it may not contain (if from the manufacturer) is information on intra- and inter-assay CVs and curve fit R2, as these are based on your data obtained from this study. If this information is not in the supplementary file, it would be good to calculate these CVs and R2s and provide them in the text. What I forgot to mention in my last review is that the number of replicates of each sample per ELISA assay should be stated as well. Did you use 2 or 3 replicates of standards and sample?

I think your information about the horses in the methods is a lot more through now, but it would be nice to know if all horses received the same amount of hay and commercial feed, or if they received feed based on body weight.

In your discussion (line 243) you state that the use of beta-glucan is safe. Further down in the discussion you say that further research is needed to comprehensively establish safety. This seems contradictory. Maybe say “based on findings of this present study beta-glucan appears safe” in line 243?

Unfortunately, you chose not to disclose your source of beta-glucan as well as its purity. I am unclear on the reason. Do you have a conflict of interest? Did the producer sponsor the research, and you did not disclose it?

Examples of language issues:

Line 45-46: “Promotes some interference in modulation of the immune response” – I am not sure what this means. “Modulates the immune response”?

Line 54: “As a result to” should be a “as a result of”

Line 57: “In response to this aggression” – I am not sure what this means. “In response to systemic LPS”?

Line 68: “The current market appeals are for alternative”. I am not sure what you mean here. “The current market asks for”?

There are many more language issues like this throughout the manuscript.

Comments on the Quality of English Language

Unfortunately, there are still language issues in this manuscript, particularly in sections added during the last review process. I have provided examples of language issues in my response to the authors. 
